# Heroic Vulnerability and the Vietnamese Refugee Experience in Thi Bui's *The Best We Could Do*

María Porras Sánchez

Department of English Studies, Complutense University of Madrid, 28040 Madrid, Spain; m.porras@ucm.es

**Abstract:** Autographics illustrating refugee and migrant experiences are frequently published, proof of the power of comics to engage with representations of trauma and vulnerability. Thi Bui's graphic memoir *The Best We Could Do* tells the story of the author and her family as "boat people", before and after migrating from Việt Nam to the US in the so-called second "wave" of refugees (1978–1980). If, as Judith Butler argues, vulnerable lives are more grievable when exposed and acknowledged, then self-representation of vulnerable lives might offer a site of resistance against precarity. Thi Bui's graphic memoir is no exception, since she deals with common themes in Vietnamese American literature such as PTSD, inherited family trauma or everyday bordering, inscribing herself and her family in the counterhistory of the US regarding the Vietnam War, while also addressing themes and motifs recurrent in Asian American comics. The author follows a thematic concern present in Vietnamese American narratives, which tends to present the refugee experience from a heroic perspective, but this is limited and antagonised by Bui's personal story, who feels estranged from her parents, their past in Việt Nam and the war. As this article shows, the recording and commemoration of her parents' memories help her to identify with the family legacy of heroic vulnerability in her role as a mother.

**Keywords:** vulnerability; autographics; Vietnam war; everyday bordering; trauma; Thi Bui; Vietnamese American literature; Asian American comics





## 1. Introduction

For Vietnamese Americans, borders are an intrinsic part of their existence; symbolised in the space that separates and unites both demonyms, they do not invoke a geographical accident—a line in a map, a stretch of land or water—but a cultural, historical and racial border.[1] Vietnamese American writer and critic Viet Thanh Nguyen summarises this division with an adversative sentence: "I was born in Vietnam but made in America" (V. T. Nguyen 2016, p. 1). This duality—origin versus destination, nature versus nurture—is marked and sustained by the war: the Vietnam War has wrought a monolithic image of the Vietnamese people in the US and has prevented them from becoming full-rights citizens. The war and its aftereffects have defined the Vietnamese American as refugees, their representation in the media and popular culture, and also their links with their ancestral homeland. This is an example of "everyday bordering" (Yuval-Davis et al. 2019), by which individuals struggle daily with the effects of borders in symbolic and tangible ways.

Vietnamese Americans are a vulnerable community on account of their race and refugee status. Hence, their lives are precarious and often affected by additional intersecting markers such as age, class, gender, or sexual identity. As Judith Butler suggests, precariousness is a common aspect of the human experience—all humans are exposed to illness, violence and death—that creates an inherent reliance among individuals (2009, p. 3). While precariousness is a fundamental aspect of existence, precarity encompasses politically induced forms of vulnerability (Butler 2009, p. 13). Butler highlights that those whose lives are not regarded as "potentially grievable, and hence valuable, are made to bear the burden of starvation, underemployment, legal disenfranchisement, and differential

exposure to violence and death" (Butler 2009, p. 25). However, vulnerable lives are more grievable when exposed and acknowledged; hence, representations of vulnerable identities can become a site of resistance against precarity.

Published in 2017, Thi Bui's *The Best We Could Do* offers such a story of resistance: Bui was, like Nguyen, born in Việt Nam[2] but made in America. Her graphic memoir tells her parents' childhood and youth in Việt Nam, their migration to the US, the family's experience as refugees in the US and Bui's own maternity. In a way, the memoir includes multiple coming-of-age stories: Bui's experience, as well as her mother's and father's, which creates a parallelism between their experiences, marked by different forms of vulnerability. However, such vulnerability is not recreated in a self-deprecating or a pessimistic fashion, but rather with a slight heroism, a frequent combination in Vietnamese fiction (V. T. Nguyen 2016, pp. 44–46).

To analyse Bui's memoir, this article describes how comics have historically addressed vulnerability and precarity and their effectiveness through their capacity to record migration and refugeeism and process traumatic memories. It also frames Bui's memoir within the thematic concerns of Asian American comics and Vietnamese American literature to show how heroic vulnerability is key to Bui's description of Vietnamese American identity and experience, and how the historical and cultural borders between Việt Nam and the US are appropriated by Bui, not only to emphasise difference but also to offer a personal history of the war that alters US hegemonic discourse.

## 2. Comics Representing Vulnerability and "Refugee Comics"

As an integral part of popular culture, comics have traditionally depicted poverty, often in connection with young characters, a trend that currently overlaps with the manifold representations of precarious lives in fictional and non-fictional comics (Porras Sánchez and Vilches 2023, p. 7). Drawing from Judith Butler (2009), who distinguishes between precariousness as an existential state of life and precarity as a political notion induced by different forms of vulnerability, Porras Sánchez and Vilches situate precarity and precariousness as recurrent motifs in contemporary comics reflecting social and individual vulnerability, systemic violence, migration, labour-based insecurity and various forms of instability (2023, p. 2). In a neoliberal context where capitalism dominates, fragmented social structures lead to fragmented lives, in what Zygmunt Bauman has defined as "liquid modernity" (Bauman 2000, 2007). Since the 1990s, new aesthetic forms have emerged to register a shift in how societal fantasies adapted to the structural pressures of crisis and loss, as Lauren Berlant has studied (Berlant 2011).

Instead of turning to retrotopias, or the nostalgic yearning for a past era or social structure perceived as more secure or stable (Bauman 2017), comics depicting precarity do not present a nostalgic portrayal to challenge the precarious present. Their graphic narratives offer an acknowledgment of existential precariousness affected by different forms of precarity, inviting readers to share and relive these experiences of vulnerability, fostering a collaborative engagement with them. Discussing the prevalent presence of precarity in the context of Asian American literature and culture, Hade Tsui-yu Lee connects it with trauma, history and ethical responsibility: "Reading historical precarity through trauma shifts the critical attention on trauma from seeing it as individual experience to seeing the traumatic symptom as a type of social structure, prompting us to redefine history and our ethical and political relation to history" (Lee 2020, p. 6). Thus, traumatic memories, when shared, go beyond the individual realm, and enter the collective dimension, contributing to revaluating history. In the case of Bui's memoir, Oh and Ninh indicate that "ethical storytelling signals to individual experiences anchored in national history and its exclusions" (2023, p. 50), highlighting its potential to engage with students in the classroom. The collaborative engagement is even more evident in the case of autographics, that is, autobiographical and autofictional comics as defined by Gillian Whitlock (2006), since the representation of the self, as in the case of Thi Bui's memoir, calls for an open identification between authors and readers. This dynamic contributes to the creation of a "precarious community"

(Claviez 2016) in which vulnerability is exposed and shared, invoking solidarity and empathy. Not in vain, the acknowledgment of vulnerability opens up an interesting possibility, since "recognition wields the power to reconstitute vulnerability" (Butler 2004, p. 43). This way, recognising vulnerability becomes a form of empowerment. By acknowledging their own vulnerability, these creators activate a mechanism of resistance and agency.

Vulnerability is recreated across a wide range of comic traditions, aesthetics and registers. These include autographics, but also coming-of-age stories, family narratives, essayistic approaches, metaphorical explorations and instances of humour, self-irony or grotesquery. Autographics can be considered especially apt when it comes to representing vulnerability and subsequent trauma. Since each individual reacts to an experience in a different way, the graphic approaches to experiences of migration, exploitation, abuse, or precarity differ greatly from one to another. If, according to Butler's ideas on precariousness, the body is vulnerable, all lives are equally perishable; the (self)representation of the body in autographics—the embodiment—always takes the centre position in these works, whether distorted, grotesque, torn, repeated, exaggerated, cute, crude, or simplified. The visualisation of such precarious bodies made them more grievable, in Butler's terminology, and their exposure is a reminder of how fragile all bodies are, even the reader's.

The possibilities of comics to address trauma and memory have been profusely studied by scholars such as Chute (2016), Romero-Jódar (2017), Nabizadeh (2020) and Davies and Rifkind (2020). The migrant and refugee experience has also been specifically addressed by comics criticism: Rifkind (2017) and Mickwitz (2020a, 2020b) use the term "refugee comics" to underscore their contrapuntal nature in challenging prevailing political and media discourses (2020b, p. 278), since they elaborate on the scant information provided by mainstream media, where refugees are reduced to figures or anonymous faces. Similarly, in *Immigrants and Comics*, Serrano argues that graphic narratives, as spaces of transaction, remembrance and mimesis,

> cast a light on what lies underneath the façade of the everyday cultural constructions of national identity [...] They testify not to a homogeneous melting pot, but rather to a multitude of contradictory and complex identities. They address and perform an accurate multiculturalism. (Serrano 2021, p. 2)

This way, comics such as Bui's offer a counternarrative to American national identity, alluding not only to the heterogeneity of the US cultural space, but also to the heterogeneity of the refugee experience.

Comics dealing with migration and refugeeism often portray situations of violence, abuse or death, which are traumatic circumstances that pose challenges to visual representations: should these experiences be presented with realistic rawness? Should they imitate the supposed objectivity of photorealism to convey the events more effectively? This is rarely the case: comics present a rich array of stylistic possibilities and visual resources that can be used to communicate harrowing themes without being explicit. In a way, comics are the art of ellipsis, a medium that works through a system of panels, pages and grids in which the action often takes place outside the page, so the reader can fill in the blanks of the story. In the productive space of the gutter—the space between panels—readers project their own recreation of events; they need to put their imagination into motion to infer what has happened. Not only gutters are a productive space for projecting the imagination. Comics often present visual metaphors and metonymies to express what cannot be recreated in a naturalistic way. Due to codification and fragmentary nature, many graphic narratives dealing with vulnerability and trauma ask for the direct participation of the reader, encouraging them to "fill in the blanks" to decode traumatic experiences, which are suggested but not explicitly recreated. It is a form of asking the reader to invest creatively and productively in the story (Nabizadeh 2020, p. 4).

In addition, according to comics critic Hillary Chute in her fundamental text *Disaster Drawn* (2016), comics that deal with trauma from an autobiographic perspective are a form of resistance in themselves: "The visualization of the ongoing procedure of self and subjectivity constructs 'ordinary' experience as relevant and political, claiming a space

in public discourse for resistance" ([Chute 2016](), p. 139). In a way, all autographics can be considered a material example of resistance. After all, they are testimonies of lives impacted by vulnerability and trauma, a materialisation of memory, a process that El Refaie defines as "commemoration" ([El Refaie 2012](), pp. 100–101). For authors of autographics, visualising vulnerability is a way of appropriating and using it to define themselves, not in a limiting way, as an imposition from the outside world—as it often happens with identity markers—but as one more layer to add in the complex apparatus that is a personal experience. It is in this commemoration that comics recording traumatic memories turn them into material examples of resistance that promote the readers' empathy, solidarity and even identification. In the following sections, I will describe how commemoration is a recurrent device in Asian American comics and Vietnamese American literature and how Bui commemorates her family stories and memories of migration.

### 3. Everyday Bordering, Family and Memory in Asian American Comics

Eleanor Ty, editor of the first volume entirely devoted to Asian American graphic narratives, underlines that Asian American comics authors have made an effort, among other issues, "to rewrite official history: re-present the everyday, unexoticized experiences of Asian Americans; intervene in collective fantasies about American heroes and superheroes" (2023, p. 2). Their need to rewrite official history responds to a need to inscribe themselves into a history that has erased, discriminated, or objectified them.

What these Asian American authors are rejecting is "everyday bordering"; according to Yuval-Davis, Wemyss and Cassady, everyday bordering refers to how individuals and groups experience and negotiate borders in their daily lives, both in tangible and symbolic terms (2019). For these authors, "borders need to be seen as constitutive of the world rather than as dividing an already made one" and "should not be viewed just as an application of top-down macro social and state policies; they are present in everyday discourses and in the practices of different social agents, from state functionaries to the media and all other members of society" ([Yuval-Davis et al. 2019](), p. 23). Through different forms of everyday bordering (e.g., recurrent racism and discrimination), individuals and groups are constantly framed as Others who do not fit in the national paradigm, carrying the border with them. Or, as Bradley and De Noronha put it, borders "follow people around, excluding them in various ways at different times, thus producing the precarity and disposability that characterizes the migrant condition" ([Bradley and De Noronha 2022](), p. 5).

In cultural representations, everyday bordering manifests itself through stereotypes; Asian Americans have been traditionally and monolithically represented through Orientalist depictions of "yellowface", as characterised by "racial" features considered "oriental", such as "'slanted' eyes, overbite, and mustard-yellow skin color" ([Lee 1999](), p. 2). In the first half of the twentieth century, there were well-known caricatures that reduced the Asian American collective to "sinister villains, exotic temptress figures, or else asexual servants with characters such as Fu Manchu, Lotus Blossom, and Hop Sing" ([Ty 2023](), p. 5), while in more recent years, the notion of the "model minority" has also stereotyped Asian Americans, condemning them to "noncitizens" and "perpetual foreigners" ([Ty 2023](), p. 5), based on a supposed identity marked by the ability to be "good citizen, productive worker, reliable consumer and member of a niche lifestyle suitable for capitalist exploitation" (V. T. [Nguyen 2002](), p. 10). Given these harmful representations, the role of graphic creators is to resist and attempt to go beyond "one-dimensional characters, racist iconography, and stereotypical representations" ([Ty 2023](), p. 5), showing that, as people involved in everyday bordering, they "construct and reconstruct the border, as well as their own identities and claims of belonging, through the creation of sociocultural, political, and geographical distinctions" ([Yuval-Davis et al. 2019](), p. 27).

The resistance against the representations of Asian Americans in US culture means offering alternative stories/representations, and numerous graphic narratives by Asian American authors deal precisely with the commemoration of migration, memory, history, identity and belonging. An early example is the first work created by an Asian person, the

first manga published in the US, *Manga Yonin Shosei* or *The Four Immigrants Manga* (1931), by Henry Kiyama. Kiyama was a Japanese migrant who recorded his experience as a student and worker in the US between 1904 and 1924. Initially conceived as a series of 52 stories and including texts in Japanese, English and Cantonese, such linguistic hybridity prevented Kiyama from publishing his stories in a Japanese magazine, so he had to resort to self-publishing and finally compiled the stories in a single volume, published in San Francisco. *Manga Yonin Shosei* revolves around racism and the everyday problems of the Japanese migrant community, but it also records important historical events such as the San Francisco earthquake in 1906 or the Universal Exhibition in 1915, mixing autobiography and history with humour. Many contemporary Asian American comics are also autobiographical: *They Called Us Enemy* (2019), by George Takei, Justin Eisinger and Steven R. Scott, narrates Takei's childhood and incarceration in an internment camp for Japanese Americans during World War II, one of the most obnoxious episodes of bordering and racial hatred committed in US soil in the second half of the 20th century. In *Vietnamerica: A Family's Journey* (2011), by GB Tran, the author retraces his family history in Việt Nam after spending all his childhood and youth oblivious to their past as refugees and his family's role in the war. In her memoir *Good Talk* (2018), Mira Jacob recreates conversations with her family in the US and India to assess migration and racism in the contemporary US, but also colourism within Indian communities, love and family. Japanese American Mari Naomi draws extensively from her biography in her numerous comics, focusing mostly on love, sex and gender, except in the memoir *Turning Japanese* (2016), in which she tells of her efforts to connect with the Japanese language and culture. Of mixed Filipino heritage and belonging to different generations and graphic styles, cartoonists Lynda Barry (*One! Hundred! Demons!* 2002) and Malaka Gharib (*I Was Their American Dream*, 2019; *It Won't Always Be Like This*, 2022) also explore their childhood and youth in Filipino migrant households, struggling with everyday bordering and the expectations imposed on them by their parents.

Other Asian American authors resort to fiction while keeping similar thematic concerns: in *American Born Chinese* (2006), Gene Luen Yang fictionalises the coming-of-age experiences of young Chinese Americans and their struggles to fit in the contemporary US, mixing everyday occurrences with Chinese mythology, even appropriating yellowface through the character of Chin-kee. Similarly, Vietnamese American Trung Le Nguyen mixes autobiography and mythology to build the story of a teenager struggling with his sexual identity who acts as a cultural and linguistic mediator for her Vietnamese-speaking mother in *The Magic Fish* (2020). However, not all Asian American authors address themes connected to their ethnic identity: Japanese American Adrian Tomine also explores autobiography, albeit his Japanese heritage occupies a marginal position in his comics, focusing instead on the role of the cartoonist and the ups and downs in his career, in the case of *The Loneliness of the Long-Distance Cartoonist* (2020).[3]

## 4. Remembrances of the Vietnam War and Heroic Vulnerability

Notwithstanding the intersection of thematic concerns in Asian American comics, such as the exploration of difficult childhoods, gender and sexuality (Ty 2023, p. 6) and the prevalent presence of the immigrant nuclear family in Asian American literature (Ninh 2011), the Vietnamese American experience has peculiarities of its own. It does not only show the evident distinctiveness of the historical and cultural background of any community, but also their representation in US culture. Vietnamese American critics agree that the war is central to the perception and representation of Vietnamese refugees in the US and also their descendants (Lee 2020; V. T. Nguyen 2016; Pelaud 2011; Oh and Ninh 2023). In the words of Berg, Việt Nam "for the U.S. had always been more a war than a country" (Berg 1986, p. 93). After twenty-four years of war, the longest in US history, and manipulative representations of the war by the mainstream cultural industry, the identification of Việt Nam with the war, and the homogenisation and depersonalisation of all Vietnamese people were imbricated in the US mindset. Nguyen stresses that if Việt Nam was equal to the Vietnam War for the US, the Vietnamese who won that war would call

it the American War (2016, p. 6). None of the names suffice, and both are "false choices", since, apart from the US involvement in many wars before and after Vietnam, the war involved other countries, other combatants and other locations (V. T. Nguyen 2016, p. 7). In any case, the war remained a site for trauma in the American imagination, according to Hade Tsui-yu Lee: "an experience of total loss" and a "despairing experience [that] acted like a 'ghost' in American society, haunting many Americans with lingering and recurring nightmares" (2020, p. 21). However, trauma mostly affected the Vietnamese refugees, displayed through "a post-traumatic stress symptom of prevailing despair, feeling disillusioned about the myth of American power" (Lee 2020, p. 21).

The arrival of South Vietnamese refugees following the victory of communist forces in 1975 caused the largest population movement to the US since the migration of Jews during and after World War II, amounting to a total of 1.5 million displaced persons, the largest population of Vietnamese people in the diaspora (Pelaud 2011, p. 8). However, their arrival was gradual, and sociologists differentiate between three different "waves": the first wave spans from 1975 to 1978, the second from 1978 to 1980 and the third from about 1980 to 1995. While the first included South Vietnamese elites and those in close contact with the US military, the third wave refers to those who arrived in the US as refugees and immigrants under the Orderly Departure Program (Pelaud 2011, p. 12). Thi Bui and her family are representatives of the second wave, including those known as "boat people", who fled Việt Nam in large numbers through dangerous sea routes to Thailand and Malaysia, where they reached refugee camps waiting for their resettlement to the US and France, mostly, a process that could last years.

Such precarious circumstances and voyages left a long-lasting impression on Vietnamese refugees, who constantly allude to war and migration in their cultural manifestations, emphasising their vulnerability. In *The Gift of Freedom* (2012), Mimi Thi Nguyen argues that Vietnamese refugees are presented with the gift of freedom, first from the war but also through their refugee status. She defines it "as an assemblage of liberal political philosophies, regimes of representation, and structures of enforcement that measure and manufacture freedom and its others (M. T. Nguyen 2012, p. 11). Such a gift comes with a demand for thankfulness and loyalty, an imposition of debt to empire (2012, pp. 2–4). In a way, vulnerability counteracts the unsolicited debt imposed by the gift, since it renders visible the emotional damage undergone in the process of reaching such freedom.

As a minority, the process of recording their vulnerability takes a heroic stance, as seen by Viet Thanh Nguyen, since, for those "dominated, excluded, exploited, or oppressed, the antiheroic takes time to develop" because "weaker populations can ill afford to seem less than powerful to the powerful" (V. T. Nguyen 2016, p. 43). Drawing from Svetlana Boym, Nguyen frames this heroic mode as a strategy of "restorative nostalgia" (V. T. Nguyen 2016, p. 43) by virtue of which less privileged individuals and groups attempt to regenerate what was lost in the past. For Vietnamese refugees, the commemoration of remembering one's own is also a self-protection device to move on from trauma, which is why, according to Nguyen, Vietnamese American art, literature and film "prefer the beautiful to the grotesque and the heroic to the antiheroic" (V. T. Nguyen 2016, p. 43), a mechanism of protection that enables refugees to adapt "to the promise of the American dream, albeit with some degree of ambivalence" (V. T. Nguyen 2016, p. 44). Additionally, the reason for this heroic stance is the need of Vietnamese refugees to claim a space for themselves in the US public sphere, to become visible, neither as victims nor as victimaries:

> For these Vietnamese exiles in America and many of their descendants, remembering one's own takes place in relationship to, and often antagonism with, the national projects of remembering one's own in Vietnam and America. These projects often ignore them and when they do notice them, usually cast them in less than heroic terms. So it is that Vietnamese Americans, for now, insist on the heroic mode in remembering themselves. (V. T. Nguyen 2016, p. 44)

Remembrance plays a key role in Vietnamese American literature and the comics of the Vietnamese diaspora. GB Tran's *Vietnamerica* shares similarities with Bui's approach in

that Tran, a member of the first wave of refugees, also explores intergenerational dynamics and reconstructs family history by interviewing different family members, although he focuses less on the storytelling process than Bui. Tran's depiction of his father is ambivalent and heroic but also distant and severe, especially towards his older children, while his mother is presented as compliant and understanding. Tran's parents' approach to survival echoes the title of Bui's memoir: "Everyone had to do whatever they needed to survive" (Tran 2011, p. 240).

As I will develop in the next section, Thi Bui's memoir fits perfectly into Nguyen's characterisation, with exceptions. While vulnerability and heroism are present in Bui's retelling of her parents' departure from Việt Nam and their adaptation to the US, such a heroic stance is antagonised by Bui's personal story, who feels estranged from her parents, their past in Việt Nam and the war. As a member of the second generation, who led a less vulnerable existence thanks to the efforts of her parents, she initially feels that her lack of heroism is her flaw. Her graphic memoir makes a triple attempt: to serve as a witness to the past by retelling her parents' stories; to close the intergenerational gap between her and her parents; and to gain visibility and define her identity as a member of the Vietnamese diaspora.

### 5. Thi Bui's *The Best We Could Do*: Heroic Vulnerability and Intergenerational Antagonism

Born in war-torn Saigon in 1975, "three months before South Việt Nam lost the war to the North" (Bui 2017, p. 39), three-year-old Thi Bui migrated by boat to Malaysia with her family, where they were granted asylum in the US. They were relocated to Hammond, Indiana, where they lived with relatives until they moved to San Diego, where the family is still based, albeit in different households. Bui's graphic memoir *The Best We Could Do* is an account of a vulnerable migrant childhood and youth marked by precarity and uncertainty, but it is also the story of her parents, Nam—called Bố, "dad", by Bui—and Hằng—called "Má", the Southern Vietnamese word for "Mom" by Bui, although Hằng prefers "Mẹ", the variant from the North—including their childhood, youth, the beginnings of their relationship, the birth of their six children, their struggles when the war started and their departure from Việt Nam.

Unlike Tran, who focuses on his father's coming of age, Bui includes herself in the diegesis and displays a triple coming of age—hers, Hằng's and Nam's—focusing on their growth from childhood to maturity, a recurrent motif in Asian American comics (Ty 2023, p. 6). At a narrative level, the three coming-of-age experiences appear as three crisscrossing storylines, intertwined through flashbacks and flashforwards, threading a complex narrative web. However, the circularity of the memoir gives it coherence, since the story begins and ends with Bui giving birth to her first son. Such a structure already points to the fact that family, parenthood, memory, migration and vulnerability are woven together in a complex network the reader needs to disentangle. The fact that Bui suffers obstetric violence, together with the challenge posed by breastfeeding, makes the experience of giving birth indelibly traumatic (Drews 2022, p. 121). And it also points to the fact that trauma frames the story and connects the different generations. The non-illustrated preface offers an insight into how the memoir was conceived and what the main theme for the author is: she started to record her family's stories in 2002 for a class project, which included some art and photographs, and then decided to turn that material into a graphic novel, her first comic, a project that took her many years.[4] Influenced by "immigration issues" and the lives of her students, she also wanted it to be a reflection of refugeehood (Bui 2017, n.p.). In 2011, coinciding with her move from New York to San Diego to be closer to her aging parents, Bui "realized that the book was about parents and children" (2017, n.p.). This intergenerational tension is the central pillar of the story, and themes such as migration, bordering, trauma and identity should be seen through that prism permeating everything.

Mirroring the precarious lives the memoir recreates, Bui's graphic style is bare and frugal, yet highly expressive, with attention to facial expressions, which are more detailed

than bodies and backgrounds. Her influences do not include Vietnamese comics, whose works are difficult to acquire in the US, but a variety of international authors such as Craig Thompson, Eleanor Davis and Jillian Tamaki and, more specifically, "Paul Pope's brush work, Aristophane's brush work, Taiyo Matsumoto's figures and composition, Gipi's washes and use of space" (Bui, qtd. in Tisserand 2017). The use of colour is reduced to black ink, the ivory of the paper, and an auburn watercolour applied with different intensities, which reminds the reader of rust or dried blood. But it mostly resembles the sepia toning applied to black-and-white photographs to create a warmer tone. The effect is tinted with nostalgia, as opening an old family photo album, even if the drawing style is quite schematic. Sometimes Bui uses darker shades of this rusty orange to intensify dramatism, and even to emphasise her father's alienation in the Communist North Việt Nam. Also, the use of a single colour might be a strategy for underlining the inextricable connection between the author, her parents, their history and her future.

However, in the last part of the memoir, we are able to see the "real" faces of the protagonists through the use of photographs resembling mugshots, taken when the family was at the refugee camp in Malaysia in 1978. A frequent presence in documentary narratives, photos convey memorialisation, a counterpoint that highlights "autobiography's claims to historical accuracy and self-reflexivity" (Chaney 2011, p. 4). But bringing up real photographs when precisely narrating the impasse in the refugee camp is also a way of achieving visibility while denouncing their objectification, as the text on the page reinforces: "We were now BOAT PEOPLE/five among hundreds of thousands of refugees flooding into neighbouring countries, seeking asylum" (Bui 2017, p. 267).

In a double page at the beginning of her memoir, Thi Bui summarises the history of Việt Nam from the first Indochina War in 1945 to the Fall of Saigon in 1975 (2017, n.p.). Following suit, her story begins with a full page of her body, ready to give birth to her first son in New York in 2005. This way, the author connects Việt Nam and the US, past and present, the historical and the familiar, the collective and the personal. In an excellent example of embodiment, Bui represents her body lying and in labour from her point of view, as if she was looking at herself, without representing the upper part (Bui 2017, p. 1). Aside from reinforcing the first-person perspective at a graphic level, she promotes identification with the reader, since, in the absence of a head, looking at her body on the page mirrors the effect of looking at oneself. The obstetric violence and fear she experiences in labour are enhanced by the fact that her mother has left her alone in the room, an action that is later justified on account of Hằng's revived trauma of giving birth to six children, two of whom died young. Bui starts her memoir by showing herself at an intimate moment of utmost vulnerability, interlacing it with crisscrossing conflicts—family, violence, intergenerational resentment, PTSD and history—which she proceeds to unpack.

The tension with her parents is partially explained by Bui's belonging to what she calls "the lame second generation" (2017, p. 29): she is caught between a place of origin that she does not remember and a place where her traditional family values clash with the norm, as seen, for instance, when her parents disown their first daughter as she moves in with her boyfriend, whereupon Hằng tries to commit suicide. Apart from the guilt of not being a "good daughter" (Bui 2017, p. 33), the gap between both generations is explained by the lack of heroism of the "assholes" of the second one (2017, p. 29), suggesting that they had an easier life compared to their parents. Bui explains that she started to record "the family history" in an attempt to understand and become closer to her parents, to negotiate her own story as a refugee, and to cope with being part of the second generation:

> thinking that if I bridged the gap between the past and the present.../I could fill the void between me and my parents. And that if I could see Việt Nam as a real place, and not a symbol of something lost.../I would see my parents as real people.../and learn to love them better. (Bui 2017, p. 36)

The author emphasises these words graphically by creating a visual analogy: Thi appears as a child with a map of Việt Nam engraved on her back, the same silhouette faintly lurking in the background of the page, mingling with the boat that took the family

out of the country and far from the war.[5] That mark is also an emotional scar, a reminder of the loss and the grief inherited from her parents. Since she cannot remember Việt Nam through her memories, she is only connected to it through inherited family trauma.

Trauma is also evoked through absence: Việt Nam is an absence, a void in Bui's back, a place that only exists now in the memories of her parents, who in turn become mediators of a place that is lost. But there are also personal losses: the children Hằng and Nam lost in Việt Nam, the firstborn, Quyền, who died of an unknown illness and lack of medical attention, and Thảo, a stillborn. "How does one recover from the loss of a child?/ How do the others compare to the memory of the lost one" (Bui 2017, pp. 56–57), Bui asks herself, on a double page with panels recreating empty streets and yards in California, evoking the void left by these losses and the impossibility of full recovery. The ghostly presence of Quyền and Thảo haunts the family and Bui, projecting "a darkness we did not understand/but could always FEEL" (Bui 2017, p. 60). Bui, who feels like a "replacement" (2017, p. 50) for the lost siblings, even includes them on a page that evokes a family tree, two shadowy figures, not human, not dead, already adults, suggesting that their ghostly selves have grown up at the same pace of the living (Bui 2017, p. 29).[6]

To heal that trauma, or at least to understand it, the memoir becomes a reversal journey, a symbolic boat trip to Vietnam, to the past and the memories and the histories of her parents, shown on the page through the longing gaze of adult Bui looking at the sea in California, followed by another full page with the flimsy boat and the roaring sea when they travel to Malaysia (Bui 2017, pp. 40–41). Water is an extended metaphor throughout the whole memoir, not only linking past and present, Việt Nam and the US, as on these two pages, but also as an element providing freedom, whether literal or symbolic: in connection with imagination, when Bui tries to evade herself from "home, the holding pen" as a child; motherhood, when her son is born; her father's survival as a child, hiding from the Viet Cong; childish joy and "an escape from regular life" (Bui 2017, p. 270), as when Hằng learns to swim in Na Trang as a child, and also Bich, Bui's older sister, in the refugee camp; but also trauma, as a liquid which connects the different family members, and that Bui intends not to pass on to her son. The allusion to refugees as "flood" in the text (Bui 2017, p. 267), a word, together with "wave", used in association with migrants in mainstream media, gives an additional connotation of the bordering yet to come.

Since the structure of the memoir is not linear, the reader gets to know Bui's problematic relationship with her parents before learning their background stories, so the reader does not empathise with them straight away. The case of Nam, Bui's father, is especially revealing. Raised in French Indochina surrounded by hunger, deprivation and violence, adult Nam is depressed, possibly suffering from post-traumatic stress disorder; he is mostly a silent figure, and Bui artistically fills his silence with the smoke of his cigarette. In a visual parallelism, she presents a silent conversation with her father in which they alternatively appear as child and adult, then both as children and, finally, both as adults, to underline the need to understand the child her father once was to understand the threatening and distant adult. After recreating his story, Bui realises that inherited trauma is a form of connection, too: "I had no idea that the terror I felt was only the long shadow of his own" (Bui 2017, p. 129). Unable to overcome the terror that made him vulnerable, Nam becomes a threatening figure and a source of fear for his younger children later. However, Bui turns him into a heroic figure as the story progresses, when he becomes the improvised helmsman of the boat that takes them to Malaysia.

In turn, Bui's mother, Hằng, had a privileged childhood, attending French schools and living in a big house in Cambodia and Na Trang. War and hardship make her a determined, resourceful woman, and she becomes the provider for the family in times of need, both in Việt Nam and the US, where she accepts earning minimum wage while studying in college at night, as her teaching degree is no longer useful. Bui devotes less space to describing her childhood in the US than to the stories of her parents, but vulnerability prevails nevertheless: memories of huddling together with more than fifteen relatives in the same house in Indiana, sleeping in the same room with the rest of her family in a

small, dark flat in California, falling ill frequently, enduring racist comments and economic precarity, etc. Contrastingly, Bui shows the pride of being, like her siblings, a gifted student. But, apart from her father's grief and her mother's tenacity, she has inherited from them the ability to survive. One day, after the explosion of some oxygen tanks in the flat underneath theirs, the family calmly leaves the house one by one after taking all their valuable documents, stored throughout the years in different folders, one for each of the family: "This—not any particular piece of Vietnamese culture—is my inheritance: the inexplicable need and extraordinary ability to RUN when the shit hits the fan. My refugee reflex" (Bui 2017, p. 305).

Even if Bui's parents did "the best they could do" (2017, p. 55) to raise their children and keep them alive, and even transmitting to them their "refugee reflex", in the face of systemic violence, that sentence equates to acknowledging their powerlessness to face the extreme circumstances surrounding them: "the best they could do" is acknowledging how precarity limits their chances of survival, no matter how hard they tried, and realising that their heroism has always been carried out at the expense of their vulnerability. At the end of the memoir, Bui and her husband Travis aspire to do the best they can with their infant son (Bui 2017, p. 311). Bui's heroism might be a response to the emotional damage left by the obstetric violence and unempathetic treatment she suffered while in labour. In a revealing full-page panel, Thi represents herself going to the incubator in the middle of the night to breastfeed her baby, born with jaundice. The same page recreates the most heroic deeds in her parents' history: her father at the rudder of the boat, her mother giving birth to her brother in the refugee camp, and her, crossing the street to the hospital, while the text reads: "The first week of parenting was the hardest week of my life, and the only time I ever felt called upon to be heroic. However much my body wanted to rest, a force pulled me onto my feet with the clear and simple directive—KEEP HIM ALIVE" (2017, p. 312). As a second-generation refugee, her inheritance to her child will be freedom from "war and loss", that is, from the traumatic and ghostly memories of the war, not survival.

## 6. Conclusions

Comics addressing vulnerability, such as *The Best We Could Do*, function at a political level and an emotional level. As ethical statements, they bear witness, expose trauma, raise empathy, appropriate vulnerability and invite readers to partake in the precarious condition, calling for an open identification between authors and readers, even if their experiences are dissimilar from the authors. This is a shared vulnerability that creates a regenerative source of identification and new forms of identification and community, an optimistic note in these liquid times. In political terms, Bui commemorates her family's stories and memories, fighting the bordering created by the monolithic image of the Vietnamese in the US, enriching the cultural construction of national identity.

Bui's memoir is inscribed within the tradition of Asian American comics and Vietnamese American narratives exploring family and coming of age, bearing witness to the past, reclaiming a space for the diaspora in the US and commemorating personal experience. Vulnerability prevails in these representations, and Bui opts to combine it with heroism. Heroism, however, does not replace vulnerability and it does not mend the past; there is just an acknowledgment of the little heroic efforts the family has made in the face of unsurmountable precarious circumstances. Bui's heroic vulnerability focuses on the role of her parents as saviours of the family at different points in the story; contrastingly, Bui's role as a mother of an ill child is compared and even superimposed onto these heroic efforts, so she ends up identifying herself with the family legacy of heroic vulnerability, the missing intergenerational link between Bui and her parents and their mechanism of resistance.

However, unlike many Vietnamese American narratives, the memoir does not promote "restorative nostalgia" (V. T. Nguyen 2016, p. 43), but rather a restorative futurity. Recording and retelling her parents' memories of Việt Nam has been a way of breaching the intergenerational gap, not a form of symbolically recovering Việt Nam. Instead, Bui projects her story to the future, reconciled with her heritage after commemorating her

parents' memories, with the hope that her son will be free from the past, free from loss and free from inherited family trauma.

**Funding:** This research received no external funding.

**Institutional Review Board Statement:** Not applicable.

**Informed Consent Statement:** Not applicable.

**Data Availability Statement:** Not applicable.

**Conflicts of Interest:** The author declares no conflict of interest.

## Notes

[1] The preliminary research for this article was carried out at Instituto Franklin (University of Alcalá) in September 2023, during a research stay funded by a Margaret Fuller Scholarship, jointly awarded by Instituto Franklin and the Association of Anglo-American Studies (AEDEAN). In this article, I use Việt Nam, the name of the country in Vietnamese, for this is the term employed by Thi Bui and other Vietnamese American authors. However, I use "Vietnam War" since the term illustrates the experience of the US in Việt Nam.

[2] In *Shortcomings* (2007), Tomine addresses Asian American identity, albeit using fictional characters.

[3] Although this article does not discuss literary works, I include several Vietnamese American titles that also tackle the Vietnam War, family memories and memory reconstruction, for instance, memoirs such as Le Ly Hayslip's *When Heaven and Earth Changed Places* (1989), wherein the author tells of her experience being tortured and rape by Vietcong combatants; Kien Nguyen's *The Unwanted* (2001), a memoir of the author's childhood growing up as a French Vietnamese kid; Jade Ngoc Quang Huynh's *South Wind Changing* (2000), Nguyen Qui Duc's *Where the Ashes Are* (1994) and Viet Thanh Nguyen's *A Man of Two Faces: A Memoir, A History, A Memorial* (2023). There are also semi-autobiographical novels, such as Andrew X. Pham's *Catfish and Mandala* (2000), lê thi diem thúy's *The Gangster We Are All Looking For* (2003) and Ocean Vuong's *On Earth We Are Briefly Gorgeous* (2019). Fictional approaches include the novel *Monkey Bridge* (1997) by Lan Cao, Aimee Phan's *We Should Never Meet* (2004), Eric Nguyen's *Things We Lost to the Water* (2021) and the novels by Viet Thanh Nguyen: *The Sympathizer* (2015) and its sequel *The Committed* (2021). For a Vietnamese French approach to the Vietnam War in the comics medium, see Marcelino Truong's *Une si jolie petite guerre—Saigon 1961–63* (2012).

[4] There are other instances of autographics of vulnerability and trauma by women authors who approached comics for the first time as adult readers and found a suitable medium to tell their stories, even if they do not come from the comics scene. This is the case of the memoirs describing sexual abuse and rape, *Becoming/Unbecoming* (2015) by British author Una or *Commute* (2018), by US author Erin Williams.

[5] In *Vietnamerica*, GB Tran also depicts the map of Việt Nam as a void, but instead of being inscribed in the body, as in Bui's memoir, is a hole in the map full of trapped people trying to escape (2011, p. 158).

[6] Ghosts of the past are also recorded as shadowy spectral presences in *Vietnamerica* (Tran 2011, pp. 65–66).

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
