# Peer review of "Heroic Vulnerability and the Vietnamese Refugee Experience in Thi Bui’s The Best We Could Do"

_humanities, doi:10.3390/h13030071_

Round 1
Reviewer 1 Report
Comments and Suggestions for Authors
Please see the attached PDF.

The quality of English is mostly good but I have highlighted in yellow on the draft numerous spots that do need work because the diction is inappropriate, usage is non-idiomatic, or the syntax doesn't make sense. Please let me know if you wish me to submit this annotated draft of the essay. I don't see an option to do it here.
Author Response
Thank you so much for your detailed reading and generous feedback. I have found your suggestions extremely useful and rich. I have included all of them. Regarding the inclusion of more allusions to Vietnamese American literary works, I have included a footnote mentioning several titles, and I have made a comment and compared Bui's work with GB Tran's Vietnamerica. I have also mentioned and commented Mimi Thi Nguyen's The Gift of Freedom. My reason for not commenting extensively on literary works is that I'd like to keep the focus on comics, which is a different medium, and a comparative approach would take too much space. I hope the footnote with the literary references palliates any possible shortcomings. I have changed the name of the section eliminating the reference to "literary" to avoid ambiguities.
With all my gratitude, all the very best.
Reviewer 2 Report
Comments and Suggestions for Authors
This was a thorough and thoughtful exploration of commemoration and futurity in Thi Bui's novel. Though as a reader I wish both terms were more deeply explained as well as signposted throughout the article, I appreciate the many insights the article gives on the text, and its contextualization within comics and Viet Nam-centered narratives.
Comments on the Quality of English LanguageMinor changes and fixes here and there were needed.
Author Response
Many thanks for your feedback, I'm so glad you appreciate my work.
All the very best
Reviewer 3 Report
Comments and Suggestions for Authors
Praise:
-The closing reading of The Best We Could Do was superb! I love the paragraphs that connect the use of the different colors/shadings in the graphic novel to connect to the precarity, and different temporal threads, that interweave in the graphic novel (lines 330 - 340)
-Smart use of expanding on the "3 waves" of South Vietnamese immigration to the US in section 4.
-"Lame second generation" and "Heroic Vulnerability" is explained well.
-I felt as though I learned a good mix of important Vietnamese American history along with seeing the impact and importance of Thi Bui's The Best We Could Do.
Possibilities:
-Honestly, the first few paragraphs were a bit hard to get through. The language and logic was a little convoluted and rushed. I think all it needs is more context on why the use of "borders" as the essential image/metaphor in the first paragraph. And then just a little more contextualizing on the use of Judith Butler in the second paragraph.
Author Response
Many thanks for your detailed reading and your generous feedback. I've tried to improve the two paragraphs in the introduction as per your suggestion, rephrasing it and including more context on Butler and borders.
All the best